# Improving Clean Accuracy via a Tangent-Space Perspective on Adversarial Training

## Abstract

Adversarial training has proven effective in improving the robustness of deep neural networks against adversarial attacks. However, this enhanced robustness often comes at the cost of a substantial drop in accuracy on clean data. In this paper, we address this limitation by introducing Tangent Direction Guided Adversarial Training (TART), a novel and theoretically well-grounded method that enhances clean accuracy by exploiting the geometry of the data manifold. We argue that adversarial examples with large components in the normal direction can overly distort the decision boundary and degrade clean accuracy. TART addresses this issue by estimating the tangent direction of adversarial examples and adaptively modulating the perturbation bound based on the norm of their tangential component. To the best of our knowledge, TART is the first adversarial defense framework that explicitly incorporates the concept of tangent space and direction into adversarial training. Extensive experiments on both synthetic and benchmark datasets demonstrate that TART consistently improves clean accuracy while maintaining robustness against adversarial attacks.

## 1 Introduction

Deep neural networks (DNNs) have achieved remarkable success in diverse domains such as computer vision (Krizhevsky et al., 2012; He et al., 2015; Yi et al., 2023; 2024), natural language processing (Otter et al., 2020), recommendation systems (He et al., 2017), and reinforcement learning (Mnih et al., 2015). However, they remain highly vulnerable to adversarial examples, which are inputs perturbed by imperceptible, malicious noise intentionally crafted to induce incorrect predictions (Szegedy et al., 2013; Goodfellow et al., 2014; Nguyen et al., 2014; Jin et al., 2020). This vulnerability poses critical security threats, particularly in safety-sensitive applications such as autonomous driving (Chen et al., 2015; Rossolini et al., 2023) and medical diagnosis (Ma et al., 2019; Finlayson et al., 2019), where even a minor error can lead to catastrophic consequences.

To address these issues, developing effective defense techniques against adversarial attacks has become a central focus in machine learning research. One of the most effective methods is *standard adversarial training* (Madry et al., 2018), which improves model robustness by incorporating adversarial examples into the training process. Following its initial success, a variety of extensions have been proposed (Gowal et al., 2020; Zhang et al., 2019; Carmon et al., 2019; Xie et al., 2018; Wong et al., 2020; Zhang et al., 2020); see Appendix B). Despite these efforts, adversarial training has consistently shown an inherent trade-off: improving robustness often leads to a notable drop in clean accuracy (Zhang et al., 2019; Yang et al., 2020; Tsipras et al., 2018). The primary goal of our study is to recover clean accuracy in adversarially trained models, without sacrificing their robustness. We approach this from a theoretically well-grounded perspective by leveraging the geometric structure of the data manifold.

While many recent studies have focused on the connection between margin and adversarial learning in the ambient space (Ding et al., 2020; Zhang et al., 2021; Wang et al., 2021), a growing body of work has begun to explore the role of data manifold geometry in adversarial defense (Lin et al., 2020; Xiao et al., 2025; Li et al., 2025). The study of adversarial examples from a manifold perspective is motivated by two key observations: first, image data lies on a low-dimensional manifold embedded in a high-dimensional input

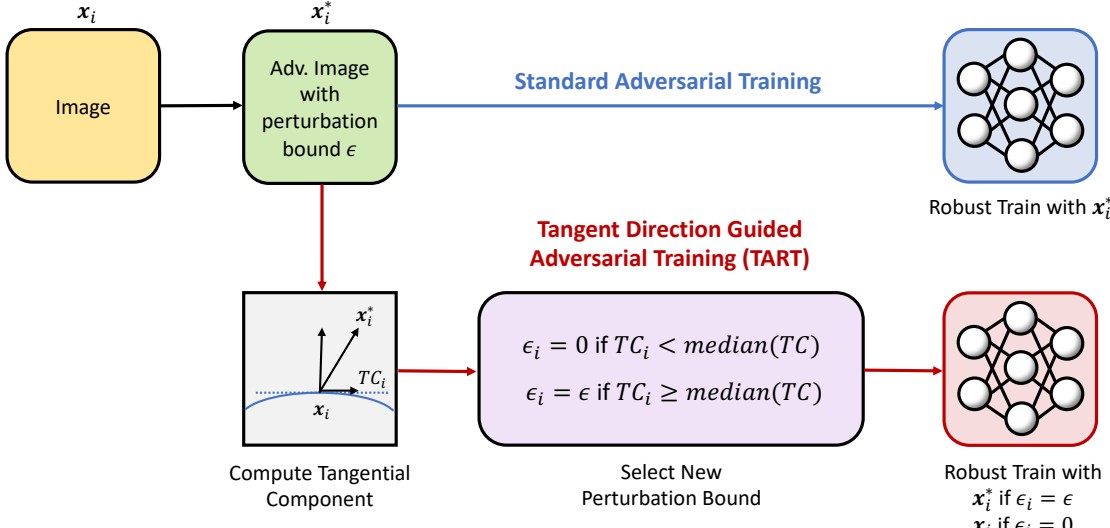

Figure 1: Overview of TART and comparison with standard adversarial training. Given a training image $\boldsymbol{x}_i$, we first generate an adversarial example $\boldsymbol{x}_i^*$ with a fixed perturbation bound $\epsilon$. Standard adversarial training trains a robust model using these examples $\boldsymbol{x}_i^*$. In contrast, TART first estimates and stores the tangent space of each training image offline using a pre-trained autoencoder and principal component analysis (PCA). Then, based on the stored tangent space information, TART computes the tangential component of $\boldsymbol{x}_i^*$. TART finally uses $\boldsymbol{x}_i^*$ for robust training if its tangential component falls within the upper 50%, and employs $\boldsymbol{x}_i$ if the tangential component is in the lower 50%.

space (Levina & Bickel, 2004; Pope et al., 2021); and second, adversarial examples are often located off this manifold (Jha et al., 2018; Tanay & Griffin, 2016; Li et al., 2021). Despite this growing interest in manifold-based approaches, to the best of our knowledge, no prior work has directly leveraged the tangent space of the data manifold to guide adversarial training.

Building on this foundation, we present a new theoretical insight into how the geometry of adversarial perturbations contributes to clean accuracy degradation in adversarial training. We formally argue that training on adversarial examples with a large normal component can significantly distort the decision boundary, ultimately leading to a substantial drop in clean accuracy. This insight has not been explored in prior work and serves as the conceptual basis for our proposed framework, **TA**ngent di**R**ection guided adversarial **T**raining (TART). TART decomposes each adversarial perturbation into its tangent and normal components and adaptively adjusts the perturbation bound based on its tangential component. By avoiding training on off-manifold examples with large normal components, TART preserves a cleaner decision boundary and achieves improved clean accuracy on unperturbed data. An overview of our approach is illustrated in Figure 1.

Our main contributions are summarized as follows:

- We introduce *Tangent Direction Guided Adversarial Training* (TART), the first adversarial training framework to explicitly leverage the tangent space and direction of adversarial examples.

- We present a theoretical analysis showing that adversarial examples with large normal components can significantly distort the decision boundary, motivating the tangent-space-guided design of TART.

- We describe a practical method for estimating the tangent space of the data manifold and computing the tangential component of adversarial examples.

- We demonstrate through extensive experiments on simulated and benchmark datasets that TART consistently boosts clean accuracy without degrading robustness. Furthermore, we illustrate that TART is a universal framework that can be seamlessly integrated with a variety of existing adversarial defense methods, further enhancing their performance.

## 2 Standard Adversarial Training

In this section, we provide an overview of standard adversarial training (Madry et al., 2018) and its implementation.

### 2.1 Notation

We focus on multiclass classification problems with $c$ classes. Let $\mathcal{D}$ be the data manifold and $\mathcal{X}$ the input feature space. The training dataset $\{(\boldsymbol{x}_i, y_i)\}_{i=1}^n$ is sampled from $\mathcal{D}$, where $\boldsymbol{x}_i \in \mathcal{X}$ and $y_i \in \mathcal{Y} = \{1, \ldots, c\}$. We assume $\mathcal{X} = \mathbb{R}^d$ and define a metric space $(\mathcal{X}, \|\cdot\|_p)$ based on the $\ell_p$ norm. Let $\mathcal{B}_p(\epsilon) = \{\boldsymbol{\delta} \in \mathcal{X} : \|\boldsymbol{\delta}\|_p \leq \epsilon\}$ denote the closed $\epsilon$-ball centered at the origin. We consider a classifier $f(\cdot; \theta) : \mathcal{X} \to \mathbb{R}^c$ parametrized by $\theta$, where the $i$-th element of the output is the score of the $i$-th class. The goal is to train a classifier $f$ that minimizes $\mathbb{E}_{(\boldsymbol{x}, y) \sim \mathcal{D}}[\ell(f(\boldsymbol{x}; \theta), y)]$, where $\ell : \mathbb{R}^c \times \mathcal{Y} \to \mathbb{R}$ is the classification loss.

### 2.2 Learning Objective

Madry et al. (2018) proposed *standard adversarial training* (standard AT), which solves a min-max optimization problem with the following objective function:

$$\min_{\boldsymbol{\theta}} \frac{1}{n} \sum_{i=1}^n \ell(f(\boldsymbol{x}_i^*; \boldsymbol{\theta}), y_i), \tag{1}$$

where

$$\boldsymbol{x}_i^* = \boldsymbol{x}_i + \arg\max_{\boldsymbol{\delta} \in \mathcal{S}} \ell(f(\boldsymbol{x}_i + \boldsymbol{\delta}; \boldsymbol{\theta}), y_i) \tag{2}$$

and $\mathcal{S}$ is the adversarial set, the set of allowed perturbations. $\boldsymbol{x}_i^*$ is also called an adversarial example of $\boldsymbol{x}$ that maximizes the loss. In the adversarial training literature, the adversarial set is typically considered to be an $l_p$ norm-bounded closed ball, i.e., $\mathcal{S} = \mathcal{B}_p(\epsilon)$. Our paper, much like other previous studies, focuses on the $l_\infty$ norm constraint, but we note that our method can be adapted for use with alternative norms as well. Standard AT improves the model's robustness by training on adversarial examples. Unfortunately, this often comes at the cost of a significant reduction in accuracy on clean data (Madry et al., 2018).

### 2.3 Projected Gradient Descent

The objective function of adversarial training involves two main steps: generating adversarial examples (inner maximization; Equation (2)) and minimizing the loss on the generated examples with respect to $\boldsymbol{\theta}$ (outer minimization; Equation (1)). The most widely used approach for the inner maximization is the *projected gradient descent* (PGD) method, introduced by Madry et al. (2018). Standard AT also employs the PGD method to obtain an approximate solution for the inner maximization problem. Given a natural data point $\boldsymbol{x}^{(0)}$, we find an adversarial example by iteratively computing the following:

$$\boldsymbol{x}^{(t+1)} \leftarrow \Pi_{\boldsymbol{x}^{(t)} + \mathcal{S}} \left( \boldsymbol{x}^{(t)} + \alpha \operatorname{sign} \left( \nabla_{\boldsymbol{x}^{(t)}} \ell(f(\boldsymbol{x}^{(t)}; \boldsymbol{\theta}), y) \right) \right) \tag{3}$$

where $\boldsymbol{x}^{(t)}$ is the adversarial example at step $t$, $y$ the class label of $\boldsymbol{x}^{(0)}$, $\Pi$ the projection operation, $\alpha$ the step size, and sign the sign function. We refer to this procedure with $k$ iterations as PGD$^k$ throughout the paper.

## 3 Theoretical Analysis of Loss Sensitivity to Normal Perturbations

In this section, we present our core theoretical analysis of adversarial perturbations. We begin by decomposing a perturbation into its tangential and normal components relative to the data manifold. We formally show that the change in loss is dominated by the normal component, while the tangential component's influence is negligible.

**Theorem 3.1.** *Let $\mathcal{M} \subset \mathbb{R}^d$ be a $k$-dimensional $C^2$ manifold, and denote by $T_x\mathcal{M}$ and $N_x\mathcal{M}$ the tangent and normal spaces at each point $x \in \mathcal{M}$, respectively. Let $f_\theta : \mathbb{R}^d \to \mathbb{R}$ be a twice continuously differentiable binary classifier parameterized by $\theta$, and let $\mathcal{L} : \mathbb{R} \times \{-1, +1\} \to \mathbb{R}_{\geq 0}$ be a twice continuously differentiable surrogate loss. For any $x \in \mathcal{M}$, consider a small perturbation $\delta$ decomposed as $\delta = \delta_T + \delta_N$, where $\delta_T \in T_x\mathcal{M}$ and $\delta_N \in N_x\mathcal{M}$, and $\|\delta\| \leq \varepsilon$ for sufficiently small $\varepsilon > 0$.*

1. *(Margin) There exists $\tau > 0$ such that $|f_\theta(x)| \geq \tau$ for every $x \in \mathcal{M}$.*

2. *(Input smoothness) $f_\theta$ is $L_x$-Lipschitz and its gradient is $\beta_x$-Lipschitz on the tubular neighborhood $\mathcal{N}_{\delta_0}(\mathcal{M})$.*

3. *(Loss smoothness) $\mathcal{L}$ is $L_z$-Lipschitz and its derivative $g(z, y) := \partial\mathcal{L}(z, y)/\partial z$ is $\beta_z$-Lipschitz in $z$.*

*Under the above assumptions, the change in loss due to tangent and normal perturbations can be characterized as:*

$$\mathcal{L}(f_\theta(x + \delta_T), y) - \mathcal{L}(f_\theta(x), y) = \mathcal{O}(\varepsilon^2),$$
$$\mathcal{L}(f_\theta(x + \delta_T + \delta_N), y) - \mathcal{L}(f_\theta(x), y) = g(f_\theta(x), y)\nabla_x f_\theta(x)^\top \delta_N + \mathcal{O}(\varepsilon^2).$$

The detailed proof for Theorem 3.1 can be found in Appendix A. This theorem provides a theoretical foundation for understanding the impact of different perturbation components on the model's loss. It shows that under a few standard assumptions, the loss change from a perturbation can be precisely characterized by its components in the tangent and normal spaces of the data manifold.

Specifically, the theorem reveals two key insights:

- A pure tangent perturbation ($\delta_T$) causes a negligible change in the loss, on the order of $\mathcal{O}(\epsilon^2)$. This is because moving along the manifold in the immediate vicinity of a data point does not significantly alter the model's output.

- The loss change from a full perturbation ($\delta_T + \delta_N$) is dominated by its normal component ($\delta_N$). The loss changes linearly with the normal perturbation, while the tangent component contributes only a higher-order term.

This theorem highlights the critical role of the normal component of adversarial perturbations in driving changes in loss. Adversarial examples with large normal components tend to push inputs far off the data manifold, into regions where the model's behavior is poorly defined. The resulting high loss in these unfamiliar regions generates large gradients during training, which can lead to significant and potentially harmful updates to model parameters. As the decision boundary shifts to accommodate these off-manifold examples, the model's ability to generalize to natural, on-manifold data degrades. These theoretical findings underscore the importance of a defense strategy that considers the geometric structure of the data manifold, particularly the tangent space.

To empirically validate our theory, we analyzed the relationship between loss and the tangential component of perturbations. We trained a VGG-16 (Simonyan & Zisserman, 2014) model on CIFAR-10 (Krizhevsky, 2009) and generated adversarial examples. As illustrated in Figure 2b, we observe a clear negative correlation: as the tangential component increases, the batch loss tends to decrease. This is because perturbations formed largely in the tangential direction are less likely to have a significant normal component, which in turn leads to a lower loss. This result supports our theoretical claim that adversarial examples aligned more closely with the tangent space are less harmful and thus help preserve clean accuracy.

## 4 Tangent Direction Guided Adversarial Training (TART)

In this section, we propose *TAngent diRection guided adversarial Training* (TART) and its implementation. TART is motivated by the theoretical findings in Section 3 that perturbations with large normal components

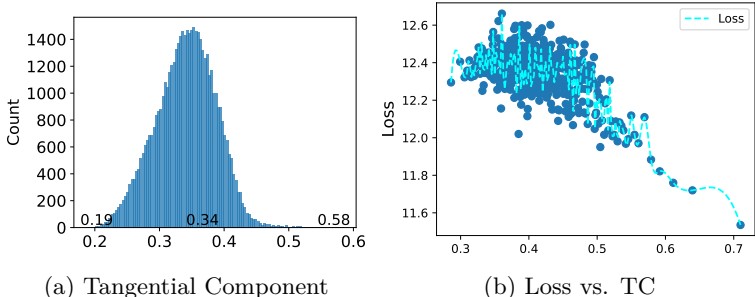

(a) Tangential Component            (b) Loss vs. TC

Figure 2: (a) Distribution of tangential components. The minimum, mean, and maximum values of the tangential component are 0.19, 0.34, and 0.58, respectively. (b) Batch loss vs. mean of tangential components (TCs) within a batch.

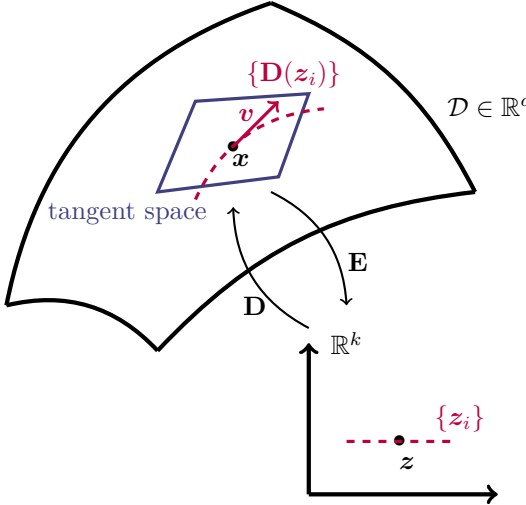

Figure 3: Illustration of Tangent Space Estimation. See Algorithm 1 for a detailed description.

can over-distort the decision boundary, ultimately reducing a model's accuracy on clean data. To address this, while standard AT uses a uniform adversarial set, TART provides a more reasonable, data-specific perturbation set that leverages information from the data manifold and tangent space.

To implement TART, we first craft an adversarial example $\boldsymbol{x}_i^*$ of $\boldsymbol{x}_i$ with the adversarial set $\mathcal{S} = \mathcal{B}_p(\epsilon)$ as described in Equation (2). Next, we estimate the tangent space at $\boldsymbol{x}_i$ and then compute the *tangential component* of $\boldsymbol{x}_i^*$, which is defined as the norm of its projection onto the tangent space. TART assigns an adaptive perturbation limit $\epsilon_i$ to each adversarial example based on its tangential component. Adversarial examples with a larger tangential component receive a larger $\epsilon_i$, while those with a smaller tangential component receive a smaller $\epsilon_i$. Using this adaptive bound, we generate the final adversarial data $\boldsymbol{x}_i^{**}$ for training:

$$\boldsymbol{x}_i^{**} = \boldsymbol{x}_i + \arg\max_{\boldsymbol{\delta} \in \mathcal{S}_i} \ell(f(\boldsymbol{x}_i + \boldsymbol{\delta}; \boldsymbol{\theta}), y_i), \tag{4}$$

where $\mathcal{S}_i = \mathcal{B}_p(\epsilon_i)$. Equation (4) can be seen as a generalization of standard AT, as it recovers Equation (2) when $\epsilon_i = \epsilon$ for all $i$.

The overall procedure of TART is illustrated in Figure 1. In the following subsections, we discuss in detail how we approximate the tangent space (Section 4.1), compute the tangential component (Section 4.2), and choose an adaptive perturbation bound used for training (Section 4.3).

---

**Algorithm 1** Finding the Tangent Space

---

**Input:** Natural data $\boldsymbol{x} \in \mathbb{R}^d$, Adversarial data $\boldsymbol{x}^* \in \mathbb{R}^d$,
       Encoder $\mathbf{E} : \mathbb{R}^d \to \mathbb{R}^k$, Decoder $\mathbf{D} : \mathbb{R}^k \to \mathbb{R}^d$
**Output:** Tangent space at $\boldsymbol{x}$

1:  $\boldsymbol{z} \leftarrow \mathbf{E}(\boldsymbol{x})$
2: **for** $i = 1, \ldots, k$ **do**
3:     Sample $\boldsymbol{z}_1, \ldots, \boldsymbol{z}_l$ around $\boldsymbol{z}$ along $i$-th dimension
4:     Obtain $\boldsymbol{X}' = (\boldsymbol{x}'_1, \boldsymbol{x}'_2, \ldots, \boldsymbol{x}'_l)$ where $\boldsymbol{x}'_j = \mathbf{D}(\boldsymbol{z}_j)$
5:     Do PCA on $\boldsymbol{X}'$ and store the first principal component $\boldsymbol{a}_i \in \mathbb{R}^d$
6: **end for**
7: Obtain $\boldsymbol{A} = (\boldsymbol{a}_1, \boldsymbol{a}_2, \ldots, \boldsymbol{a}_k) \in \mathbb{R}^{d \times k}$ where the $k$-dimensional tangent space at $\boldsymbol{x}$ is the column space of
   $\boldsymbol{A}$

---

### 4.1 Finding the Tangent Space

For datasets where the data manifold is known, the tangent space can be obtained explicitly and used directly in TART. However, for benchmark or real-world datasets, the corresponding data manifold is often unknown, and we must therefore estimate the tangent space.

Suppose a dataset $\{\boldsymbol{x}\} \subset \mathbb{R}^d$ lies on a $k$-dimensional manifold embedded in $\mathbb{R}^d$, where $k < d$. Indeed, many studies have demonstrated that the intrinsic dimension of image datasets is considerably smaller than their pixel space dimension (Levina & Bickel, 2004; Pope et al., 2021). Let $\mathbf{D} : \mathbb{R}^k \to \mathbb{R}^d$ be a parameterization of this $k$-dimensional manifold. For a data point $\boldsymbol{x} \in \mathbb{R}^d$, let $\boldsymbol{z} \in \mathbb{R}^k$ be its corresponding parameterization, such that $\mathbf{D}(\boldsymbol{z}) = \boldsymbol{x}$. By a first-order Taylor approximation, we can approximate the manifold locally as a hyperplane:

$$\mathbf{D}(\boldsymbol{z} + \boldsymbol{\delta}) \approx \mathbf{D}(\boldsymbol{z}) + \mathbf{J}_{\mathbf{D}}(\boldsymbol{z})\boldsymbol{\delta} \tag{5}$$

where $\boldsymbol{\delta} \in \mathbb{R}^k$ and $\mathbf{J}_{\mathbf{D}}$ is the Jacobian matrix of $\mathbf{D}$. Therefore, a tangent vector at $\boldsymbol{x}$ can be approximated by $\mathbf{D}(\boldsymbol{z} + \delta \boldsymbol{e}_i) - \mathbf{D}(\boldsymbol{z})$, where $\boldsymbol{e}_i \in \mathbb{R}^k$ is a unit vector. To obtain a better approximation of the tangent vector, we can sample multiple points $\{\boldsymbol{z} + \delta \boldsymbol{e}_i\}$ around $\boldsymbol{z}$ along each dimension $i$ with different $\delta$s. We then perform PCA on these sampled points and use the first principal component as an approximation of the $i$-th tangent vector. By repeating this procedure for each dimension $i = 1, ..., k$, we obtain a set of $k$ tangent vectors that span the tangent space at $\boldsymbol{x}$.

Unfortunately, the true intrinsic data manifold is unknown for most image datasets. Therefore, before approximating the tangent space, we must first learn a parameterization of the manifold, denoted as $\mathbf{D}$. A common approach is to use an autoencoder trained to reconstruct the input data (Bank et al., 2020). Refer to Algorithm 1 for the steps estimating the tangent space, and see Figure 3 for a visual representation.

### 4.2 Computing Tangential Component

Using the estimated tangent space, we compute the tangential component of an adversarial example. Let $\boldsymbol{A}$ be a $d \times k$ matrix whose columns are the tangent vectors of a natural data $\boldsymbol{x} \in \mathbb{R}^d$. We note that the column space of $\boldsymbol{A}$ represents the $k$-dimensional tangent space at $\boldsymbol{x}$. Given an adversarial example $\boldsymbol{x}^*$ obtained by perturbing $\boldsymbol{x}$, we define $\boldsymbol{w}$ as the projection of the perturbation $\boldsymbol{x}^* - \boldsymbol{x}$ onto the tangent space. This projection can be computed using the projection matrix $\Pi_{\boldsymbol{A}} = \boldsymbol{A}(\boldsymbol{A}^\intercal \boldsymbol{A})^{-1}\boldsymbol{A}^\intercal$. A more detailed explanation of this projection is provided in Appendix C. Therefore, the tangential component of $\boldsymbol{x}^*$ is calculated as:

$$\begin{aligned}
\|\boldsymbol{w}\| &= \|\Pi_{\boldsymbol{A}}(\boldsymbol{x}^* - \boldsymbol{x})\| \\
&= \left\|\boldsymbol{A}(\boldsymbol{A}^\intercal \boldsymbol{A})^{-1}\boldsymbol{A}^\intercal(\boldsymbol{x}^* - \boldsymbol{x})\right\|
\end{aligned} \tag{6}$$

**Saving Time Complexity of TART**   TART requires the estimation of the tangent space and the matrix calculation $\boldsymbol{A}(\boldsymbol{A}^\intercal \boldsymbol{A})^{-1}\boldsymbol{A}^\intercal$ for each training data point in every epoch. To reduce this significant computational overhead, we pre-compute the tangent space and the corresponding matrix $\boldsymbol{A}(\boldsymbol{A}^\intercal \boldsymbol{A})^{-1}\boldsymbol{A}^\intercal$ for

---

**Algorithm 2** TART

---

**Input:** Training dataset $\{(\boldsymbol{x}_i, y_i)\}_{i=1}^n$, Perturbation $\epsilon_{\max}$
**Output:** Trained deep network $f(\boldsymbol{x}; \boldsymbol{\theta})$

1: Train Autoencoder $(\mathbf{E}, \mathbf{D})$ on the training set
2: Compute tangent space of each training data using Algorithm 1 and store them for line 8
3: **for** epoch $= 1, \ldots, T$ **do**
4:     **for** mini-batch $= 1, \ldots, M$ **do**
5:         Sample a mini-batch $\mathcal{B} = \{(\boldsymbol{x}_i, y_i)\}_{i=1}^m$ from $\mathcal{D}$
6:         **for** $i = 1, \ldots, m$ **do**
7:             Generate adversarial example $\boldsymbol{x}_i^*$ of $\boldsymbol{x}_i$ with perturbation bound of $\epsilon_{\max}$
8:             Calculate tangential component of $\boldsymbol{x}_i^*$ by Equation (6) using the stored tangent space information in line 2
9:         **end for**
10:         Assign $\epsilon_i$ according to Equation (7)
11:         $\boldsymbol{x}_i^{**} = \boldsymbol{x}_i^*$ if $\epsilon_i = \epsilon_{\max}$
            $\boldsymbol{x}_i^{**} = \boldsymbol{x}_i$ if $\epsilon_i = 0$ for all $1 \leq i \leq m$
12:         $\boldsymbol{\theta} \leftarrow \boldsymbol{\theta} - \eta \nabla_{\boldsymbol{\theta}} \left\{ \sum_{i=1}^m \frac{1}{m} l(f(\boldsymbol{x}_i^{**}), y_i) \right\}$
13:     **end for**
14: **end for**

---

each training image before training begins and store them. One limitation of pre-saving matrices is that it restricts the use of data augmentation techniques such as random cropping or horizontal flips, as such operations would randomly alter the data manifold.

As a result of this strategy, we were able to minimize the computational burden during training. TART's training time only increased by a factor of 1.24 compared to standard AT, which is a very reasonable overhead in line with other advanced defense techniques (Zhang et al., 2021; 2019; Wang et al., 2020). For details on training time measurements and the experimental setup, see Appendix E.

**Saving Space Complexity of TART** Storing matrices to reduce training time requires a considerable amount of storage space. For example, the train set of CIFAR-10 consists of 50,000 RGB color images, each with a dimension $d = 3{,}072$. Consequently, the matrix $\boldsymbol{A}(\boldsymbol{A}^\intercal \boldsymbol{A})^{-1}\boldsymbol{A}^\intercal$ becomes a large $d \times d$ dense matrix. To address this, instead of saving a $d \times d$ matrix $\boldsymbol{A}(\boldsymbol{A}^\intercal \boldsymbol{A})^{-1}\boldsymbol{A}^\intercal$, we suggest storing two $d \times k$ matrices $\boldsymbol{A}(\boldsymbol{A}^\intercal \boldsymbol{A})^{-1}$ and $\boldsymbol{A}$. This provides significant space efficiency, as the intrinsic dimension $k$ is considerably smaller than the pixel space dimension $d$ ($k \ll d$).

### 4.3 Selecting the Perturbation Bound $\epsilon$

The final step in realizing TART is to select an adaptive perturbation bound, $\epsilon_i$, for each data during training. We first generate adversarial examples with a perturbation bound of $\epsilon_{\max}$ for each training data. Next, the tangential components computed in Section 4.2 are used as the criterion for choosing its final $\epsilon$. The fundamental idea is to assign larger or smaller $\epsilon$ values to data with larger or smaller tangential components. While several assignment methods exist, TART adopts a straightforward approach providing $\epsilon_{\max}$ to the upper 50% and 0 to the lower 50%. A brief overview of alternative assignment strategies and their results is provided in Appendix H. The perturbation bound of the $i$-th data in a mini-batch can thus be expressed as:

$$\epsilon_i = \mathbb{I}\left(TC_i \geq \text{median}\{TC_k\}_{k=1}^B\right) \cdot \epsilon_{\max} \tag{7}$$

where $TC_i$ is the tangential component of the $i$-th data and $B$ is the mini-batch size. The main advantage of this simple assignment is that it avoids the need to regenerate adversarial images. When $0 < \epsilon_i < \epsilon_{\max}$, a new adversarial example must be created with the new bound $\epsilon_i$. However, by only using bounds of 0 or $\epsilon_{\max}$, we can reuse existing images: the original image for a bound of 0, and the pre-computed adversarial image for a bound of $\epsilon_{\max}$. This significantly reduces the training time, as generating adversarial examples is computationally expensive. The overall training process of TART is summarized in Algorithm 2.

| Method | Accuracy | $\mathbb{R}^{100}$ $(c=4)$ | | $\mathbb{R}^{100}$ $(c=8)$ | | $\mathbb{R}^{100}$ $(c=16)$ | |
|---|---|---|---|---|---|---|---|
| | | $\epsilon = 0.03$ | $\epsilon = 0.05$ | $\epsilon = 0.03$ | $\epsilon = 0.05$ | $\epsilon = 0.01$ | $\epsilon = 0.03$ |
| TART | Clean | **95.3 ± 0.8** | **93.5 ± 1.5** | **93.2 ± 2.1** | **88.2 ± 2.5** | **94.0 ± 0.8** | **85.6 ± 1.7** |
| | Robust | **63.1 ± 2.6** | **48.1 ± 3.8** | **40.6 ± 1.4** | **24.8 ± 1.6** | **49.3 ± 2.4** | **18.0 ± 1.8** |
| Reverse-TART | Clean | 91.9 ± 3.7 | 87.9 ± 4.8 | 87.9 ± 1.7 | 82.5 ± 1.3 | 89.4 ± 1.3 | 82.2 ± 3.4 |
| | Robust | 57.5 ± 2.1 | 43.9 ± 3.7 | 39.8 ± 1.1 | 23.5 ± 2.3 | 46.9 ± 2.5 | 15.4 ± 1.9 |
| Method | Accuracy | $\mathbb{R}^{200}$ $(c=4)$ | | $\mathbb{R}^{200}$ $(c=8)$ | | $\mathbb{R}^{200}$ $(c=16)$ | |
| | | $\epsilon = 0.03$ | $\epsilon = 0.05$ | $\epsilon = 0.03$ | $\epsilon = 0.05$ | $\epsilon = 0.01$ | $\epsilon = 0.03$ |
| TART | Clean | **95.9 ± 1.4** | **90.7 ± 1.0** | **91.4 ± 1.6** | **88.0 ± 1.0** | **93.2 ± 1.6** | **85.4 ± 1.8** |
| | Robust | **54.7 ± 2.2** | **37.9 ± 4.1** | **29.5 ± 3.8** | **16.5 ± 1.6** | **39.3 ± 0.5** | **9.4 ± 0.9** |
| Reverse-TART | Clean | 90.1 ± 3.0 | 87.2 ± 2.0 | 88.0 ± 1.7 | 78.1 ± 2.6 | 88.9 ± 1.1 | 77.8 ± 2.8 |
| | Robust | 50.4 ± 3.1 | 37.6 ± 3.2 | 28.3 ± 2.9 | 12.7 ± 3.6 | 38.8 ± 1.8 | 7.9 ± 0.7 |
| Method | Accuracy | $\mathbb{R}^{400}$ $(c=4)$ | | $\mathbb{R}^{400}$ $(c=8)$ | | $\mathbb{R}^{400}$ $(c=16)$ | |
| | | $\epsilon = 0.01$ | $\epsilon = 0.03$ | $\epsilon = 0.01$ | $\epsilon = 0.03$ | $\epsilon = 0.01$ | $\epsilon = 0.03$ |
| TART | Clean | **97.8 ± 0.3** | **94.1 ± 1.4** | **96.9 ± 0.5** | **86.9 ± 0.8** | **91.4 ± 1.2** | **82.1 ± 4.0** |
| | Robust | **74.5 ± 0.6** | **46.5 ± 1.8** | **54.1 ± 0.5** | **18.3 ± 1.5** | **26.3 ± 0.6** | **4.4 ± 0.9** |
| Reverse-TART | Clean | 97.6 ± 0.7 | 84.5 ± 3.8 | 91.7 ± 0.7 | 82.9 ± 1.3 | 84.9 ± 1.7 | 70.5 ± 3.8 |
| | Robust | 60.6 ± 1.6 | 41.2 ± 1.5 | 50.9 ± 1.4 | 17.3 ± 2.9 | 24.9 ± 1.6 | 4.0 ± 1.1 |

Table 1: Test accuracies (%) on transformed hemisphere dataset under different dimensions $d$, number of classes $c$, and perturbation bounds $\epsilon$. The average clean and robust accuracies over five trials are reported along with their standard deviation.

# 5  Experiments

In this section, we present a series of experiments to validate the effectiveness of our proposed method, TART. We begin with a simulated experiment on a transformed hemisphere dataset, which serves to illustrate and validate our core theoretical insights. We then evaluate TART on the CIFAR-10 dataset, demonstrating that it not only improves clean accuracy but can also be effectively combined with existing adversarial training methods. Finally, we evaluate TART on the Tiny ImageNet dataset to assess its ability to generalize to more challenging and large-scale settings.

## 5.1  Transformed Hemisphere

To compare the efficacy of training on adversarial examples with large normal components versus those with large tangential components, we conducted an experiment using a simulated dataset. Our dataset is based on a unit hemisphere in $\mathbb{R}^3$, where the tangent space can be computed without any approximation. The hemisphere is first evenly divided into $c$ regions to serve as class labels.

Data points are sampled from the hemispherical surface. Let $\boldsymbol{z} \in \mathbb{R}^3$ be a sampled point, which is then transformed to a high-dimensional space $\mathbb{R}^d$ via a linear transformation, $\boldsymbol{x} = T(\boldsymbol{z})$. Here, $T : \mathbb{R}^3 \to \mathbb{R}^d$ is a linear map whose columns are three orthonormal vectors in $\mathbb{R}^d$. The class label of $\boldsymbol{x}$ is determined by the region where $\boldsymbol{z}$ is located. Since the data points lie on a spherical manifold, we can explicitly compute the tangent vectors, $\boldsymbol{u}_1$ and $\boldsymbol{u}_2$, of $\boldsymbol{z}$. The corresponding tangent vectors for $\boldsymbol{x}$ are then obtained by applying the transformation $T$ to $\boldsymbol{u}_1$ and $\boldsymbol{u}_2$. Using these tangent vectors, we can accurately calculate the tangential component of adversarial examples and implement TART on the simulated dataset.

The goal of this experiment is to confirm the validity of our core concepts and assess the effectiveness of TART. To this end, we compare TART to a contrasting approach, which we call Reverse-TART. Reverse-TART provides a smaller perturbation bound to adversarial examples with larger tangential components, directly opposing the core hypothesis of our method. To obtain a more pronounced effect, we compute

Table 2: Test accuracies (%) on CIFAR-10. The performances over three trials are reported along with their standard deviation.

| Defense | Clean (Last) | Clean (Best) | FGSM | PGD$^{20}$ | PGD$^{40}$ | AutoAttack |
|---|---|---|---|---|---|---|
| Clean | $90.72 \pm 0.10$ | $90.86 \pm 0.10$ | $5.19 \pm 1.03$ | $0.0 \pm 0.0$ | $0.0 \pm 0.0$ | $0.0 \pm 0.0$ |
| AT | $81.53 \pm 0.21$ | $81.76 \pm 0.22$ | $54.11 \pm 0.37$ | $37.06 \pm 0.27$ | $36.73 \pm 0.20$ | $35.89 \pm 0.19$ |
| TART-AT | $\mathbf{83.47 \pm 0.22}$ | $\mathbf{83.73 \pm 0.15}$ | $\mathbf{55.27 \pm 0.25}$ | $\mathbf{37.72 \pm 0.03}$ | $\mathbf{37.41 \pm 0.13}$ | $\mathbf{36.58 \pm 0.08}$ |
| TRADES | $79.47 \pm 0.09$ | $79.85 \pm 0.18$ | $54.53 \pm 0.10$ | $39.08 \pm 0.06$ | $38.68 \pm 0.06$ | $37.80 \pm 0.07$ |
| TART-TRADES | $\mathbf{81.60 \pm 0.06}$ | $\mathbf{81.95 \pm 0.08}$ | $\mathbf{55.49 \pm 0.47}$ | $\mathbf{39.30 \pm 0.26}$ | $\mathbf{38.99 \pm 0.27}$ | $\mathbf{38.26 \pm 0.30}$ |
| MART | $79.82 \pm 0.39$ | $80.18 \pm 0.22$ | $53.15 \pm 0.17$ | $36.76 \pm 0.39$ | $36.16 \pm 0.44$ | $34.28 \pm 0.56$ |
| TART-MART | $\mathbf{82.33 \pm 0.28}$ | $\mathbf{84.31 \pm 0.15}$ | $\mathbf{55.19 \pm 0.12}$ | $\mathbf{37.64 \pm 0.25}$ | $\mathbf{37.17 \pm 0.25}$ | $\mathbf{35.86 \pm 0.16}$ |
| GAIRAT | $81.37 \pm 0.15$ | $81.67 \pm 0.12$ | $54.13 \pm 0.13$ | $37.29 \pm 0.23$ | $36.89 \pm 0.25$ | $36.03 \pm 0.24$ |
| TART-GAIRAT | $\mathbf{83.14 \pm 0.15}$ | $\mathbf{83.47 \pm 0.20}$ | $\mathbf{55.24 \pm 0.38}$ | $\mathbf{37.43 \pm 0.26}$ | $\mathbf{37.04 \pm 0.32}$ | $\mathbf{36.26 \pm 0.36}$ |

Table 3: Test accuracies (%) on Tiny ImageNet. The average performances over three trials are reported.

| Defense | Clean (Last) | Clean (Best) | FGSM | PGD$^{20}$ | PGD$^{40}$ | AutoAttack |
|---|---|---|---|---|---|---|
| Clean | $55.97 \pm 0.21$ | $58.01 \pm 0.11$ | $13.58 \pm 0.35$ | $7.50 \pm 0.29$ | $6.05 \pm 0.40$ | $0.32 \pm 0.02$ |
| AT | $49.94 \pm 0.19$ | $53.18 \pm 0.53$ | $33.41 \pm 0.21$ | $26.50 \pm 0.16$ | $25.83 \pm 0.16$ | $\mathbf{25.34 \pm 0.18}$ |
| TART-AT | $\mathbf{53.19 \pm 0.07}$ | $\mathbf{56.04 \pm 0.20}$ | $\mathbf{33.74 \pm 0.09}$ | $\mathbf{26.56 \pm 0.26}$ | $\mathbf{25.87 \pm 0.28}$ | $25.11 \pm 0.08$ |

the tangential components at each epoch and use only a subset of the adversarial examples: those with the largest 25% and the smallest 25% tangential components. The perturbation bound for each adversarial example during training is determined as follows:

- TART: provide $\epsilon$ for the largest 25% and 0 for the smallest 25% as their perturbation bounds.

- Reverse-TART: provide 0 for the largest 25% and $\epsilon$ for the smallest 25% as their perturbation bounds.

We train a two hidden layer DNN across various experimental settings: dimension $d \in \{100, 200, 400\}$, number of classes $c \in \{4, 8, 16\}$, maximum perturbation $\epsilon \in \{0.01, 0.03, 0.05\}$. The models are trained for 50 epochs using stochastic gradient descent (SGD) with a momentum of 0.9 and a weight decay of 0.0002. We use an initial learning rate of 0.1, which is decayed by a factor of 10 at epochs 30 and 45. The adversarial examples for training are generated by PGD$^{10}$ with a random start and a step size of $\alpha = \epsilon/4$. Note that a perturbation bound of 0 implies the use of natural images without any PGD attacks.

In Table 1, we compare the performance of TART and Reverse-TART based on clean and robust test accuracy. Clean accuracy is the model's performance on the test dataset without attacks, while robust accuracy is measured against adversarial data generated by PGD$^{20}$ with a random start, an $l_\infty$ perturbation bound of $\epsilon$, and a step size of $\alpha = \epsilon/10$. Our results show that TART consistently outperforms Reverse-TART in both clean and robust accuracy across all cases. These findings support our core hypothesis that avoiding training on adversarial examples with large normal components is an effective strategy for enhancing clean accuracy without sacrificing robustness.

### 5.2 Performance Evaluation on CIFAR

In this section, we assess the performance of TART on CIFAR-10 (Krizhevsky, 2009). We also verify the versatility of TART by successfully integrating it with other existing defense methods. Specifically, we combine TART with the following defense approaches: (1) Standard AT (Madry et al., 2018), (2) TRADES (Zhang et al., 2019), (3) MART (Wang et al., 2020), and (4) GAIRAT (Zhang et al., 2021).

### 5.2.1 Defense Settings

We use WideResNet-32-10 (Zagoruyko & Komodakis, 2016) (WRN-32-10) as in GAIRAT (Zhang et al., 2021), FAT (Zhang et al., 2020), and MAIL (Wang et al., 2021). We train each defense model for 100 epochs using SGD with momentum 0.9, weight decay 0.0002. We initially use a learning rate of 0.1 and divide it by 10 at the 60th and 90th epochs. For the training attack, we use the $PGD^7$ attack with a random start, $l_\infty$ perturbation bound $\epsilon_{\max} = 8/255$, and step size $\alpha = 2/255$. The hyperparameters featured in each method are configured to match those used in the original paper: $\beta = 6$ for TRADES, $\lambda = 5$ for MART. Also, only for MART, we followed using a maximum learning rate of 0.01 and weight decay of 0.0035 as in Wang et al. (2020). Note that data augmentation techniques were not used during the experiment. All experiments were run on a single GeForce GTX 1080 GPU.

### 5.2.2 Evaluation Metrics

We report both the clean test accuracy at the best checkpoint and the last checkpoint. We also evaluate the robustness of all trained models against four types of adversarial attacks: Fast Gradient Sign Method (FGSM) (Goodfellow et al., 2014), $PGD^{20}$, $PGD^{40}$ (Madry et al., 2018), and AutoAttack (Croce & Hein, 2020). All attacks are constrained to $l_\infty$ perturbations with a maximum bound of $\epsilon_{\max} = 8/255$, i.e., $\|x - x^*\|_\infty \leq 8/255$. The step size for $PGD^{20}$ and $PGD^{40}$ is set to $\alpha = 2/255$, while FGSM is equivalent to a one-step PGD with step size $\epsilon$. FGSM, $PGD^{20}$, and $PGD^{40}$ attacks are conducted under a white-box setting where the attacker has full access to model parameters. AutoAttack is a combination of four different attack methods that include a black-box attack.

One major difference from the transformed hemisphere experiment discussed in Section 5.1 is the lack of knowledge regarding the exact tangent space of the data manifold. As described in Section 4, we first train a convolutional autoencoder on CIFAR-10 to approximate the tangent space and compute the tangential components. See Appendix G for more details on the autoencoder. We leverage this knowledge to evaluate the performance of TART, as well as TART combined with well-known defense methods. Our experimental results are summarized in Table 2. The results indicate that when TART is combined with existing methods, clean accuracy is consistently boosted, while robustness is maintained with no degradation. This again confirms the effectiveness of TART and suggests that examining the tangential component is a valuable approach for improving adversarial training.

For a more detailed analysis, we also performed ablation studies on the perturbation bound assignment strategies and the autoencoder latent dimension. The results of these experiments are provided in Appendices G and H.

### 5.3 Performance Evaluation on Tiny ImageNet

To further validate the effectiveness of TART on more complex datasets, we conducted experiments on Tiny ImageNet (Le & Yang, 2015). The full experimental setup, including the model architecture and training hyperparameters, is detailed in Appendix F.

As shown in Table 3, our results on Tiny ImageNet are consistent with those observed on CIFAR-10. Compared to standard AT, TART achieves higher clean accuracy while maintaining comparable robustness. These findings suggest that TART's ability to improve clean accuracy extends effectively to more complex and large-scale image classification tasks.

## 6 Conclusion

In this paper, we introduced Tangent Direction Guided Adversarial Training (TART), a novel and theoretically grounded framework that leverages the tangent space of the data manifold to guide adversarial training. Our analysis revealed that adversarial examples with large normal components can excessively distort the decision boundary, thereby impairing generalization to natural data. TART mitigates this issue by adaptively modulating the perturbation bound based on the tangential component of each adversarial

example. In particular, it assigns larger or smaller perturbation limits to adversarial examples with larger or smaller tangential components, respectively.

Through experiments on both synthetic and benchmark datasets, we demonstrated that TART effectively improves clean accuracy while maintaining robustness comparable to standard adversarial training. To the best of our knowledge, TART is the first framework to incorporate tangent directions into adversarial learning. In future work, we plan to explore sample-wise optimal perturbation bounds by theoretically analyzing each example's contribution to overall model performance. We believe that this tangent-space perspective opens new directions for geometry-aware adversarial defenses and provides a promising foundation for future research.

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

# A   Analysis of Theorem 3.1

Before proving the main result, we establish a key geometric lemma that ensures the gradient of the classifier is orthogonal to the tangent space of the data manifold.

**Lemma 1.** *Let $\mathcal{M} \subset \mathbb{R}^d$ be a $k$-dimensional $C^2$ manifold with tangent space $T_x\mathcal{M}$ and normal space $N_x\mathcal{M}$ defined at each point $x \in \mathcal{M}$. Then, for every $x \in \mathcal{M}$, the input gradient $\nabla_x f_\theta(x)$ is orthogonal to the tangent space $T_x\mathcal{M}$, that is,*

$$\nabla_x f_\theta(x) \perp T_x\mathcal{M}.$$

*Proof.* Fix $x \in \mathcal{M}$ and suppose, for contradiction, that there exists a unit vector $v \in T_x\mathcal{M}$ such that $\alpha := \nabla_x f_\theta(x)^\top v \neq 0$. Without loss of generality, we assume $f_\theta(x) \geq \tau$.

Since $\mathcal{M}$ is a $C^2$ manifold, the exponential map $\exp_x : T_x\mathcal{M} \to \mathcal{M}$ is well-defined. Define the $C^1$ curve $c(t) := \exp_x(tv)$ for $t \in (-t_0, t_0)$ such that $c(0) = x$, $c'(0) = v$, and $c(t) \in \mathcal{M}$ for sufficiently small $|t|$.

Because $f_\theta$ is $C^2$, we can apply the second-order Taylor expansion of $f_\theta(c(t))$ around $t = 0$:

$$f_\theta(c(t)) = f_\theta(x) + \alpha t + R(t), \tag{8}$$

where $R(t)$ denotes the second-order remainder term, which satisfies

$$R(t) = \frac{1}{2}(f_\theta \circ c)''(\xi)t^2 \quad \text{for some } \xi \in (0, t).$$

To bound the remainder, we compute:

$$(f_\theta \circ c)''(t) = \frac{d^2}{dt^2}f_\theta(c(t)) = c'(t)^\top \nabla^2 f_\theta(c(t))c'(t) + \nabla f_\theta(c(t))^\top c''(t).$$

From the smoothness assumptions, we have:

$$\|\nabla_x^2 f_\theta(x)\| \leq \beta_x \quad \text{and} \quad \|\nabla_x f_\theta(x)\| \leq L_x.$$

Furthermore, since $c(t)$ is a unit-speed geodesic on the $C^2$ submanifold $\mathcal{M}$, we have $\|c'(t)\| = 1$, and its acceleration is bounded by a constant $\kappa > 0$ such that $\|c''(t)\| \leq \kappa$. Using this, we can bound the second derivative of the composition:

$$|(f_\theta \circ c)''(t)| \leq \beta_x \|c'(t)\|^2 + L_x \|c''(t)\| \leq \beta_x + L_x \kappa.$$

Hence, the remainder term satisfies:

$$|R(t)| \leq \frac{1}{2}(\beta_x + L_x \kappa)t^2.$$

Now, since $f_\theta(x) \geq \tau$ and $\alpha \neq 0$, define

$$t_\star := \frac{f_\theta(x) - \frac{\tau}{2}}{|\alpha|} > 0.$$

Let $t := \text{sign}(-\alpha) \cdot t_\star$. Then,

$$\alpha t = -\left(f_\theta(x) - \frac{\tau}{2}\right).$$

Substituting this into equation 8, we obtain:

$$f_\theta(c(t)) = f_\theta(x) + \alpha t + R(t) = \frac{\tau}{2} + R(t).$$

Taking absolute values yields:

$$|f_\theta(c(t))| \leq \frac{\tau}{2} + |R(t)|. \tag{9}$$

Since we can shrink $t_0$ so that $|t| < t_0$ and make the remainder sufficiently small, choose $t_0$ such that:

$$|R(t)| \leq \frac{\tau}{4}. \tag{10}$$

Combining equation 9 and equation 10, we conclude:

$$|f_\theta(c(t))| \leq \frac{\tau}{2} + \frac{\tau}{4} = \frac{3\tau}{4} < \tau.$$

This contradicts the margin condition (A1), since $c(t) \in \mathcal{M}$ by construction, yet $|f_\theta(c(t))| < \tau$. Hence, our assumption $\alpha \neq 0$ must be false, which implies:

$$\nabla_x f_\theta(x)^\top v = 0 \quad \text{for all } v \in T_x\mathcal{M}.$$

That is, $\nabla_x f_\theta(x) \perp T_x\mathcal{M}$, as desired. □

We now proceed to prove the main theorem. The orthogonality property established in Lemma 1 will play a crucial role in decoupling the effect of tangent and normal perturbations in the loss expansion.

**Theorem 3.1.** *Let $\mathcal{M} \subset \mathbb{R}^d$ be a $k$-dimensional $C^2$ manifold, and denote by $T_x\mathcal{M}$ and $N_x\mathcal{M}$ the tangent and normal spaces at each point $x \in \mathcal{M}$, respectively. Let $f_\theta : \mathbb{R}^d \to \mathbb{R}$ be a twice continuously differentiable binary classifier parameterized by $\theta$, and let $\mathcal{L} : \mathbb{R} \times \{-1, +1\} \to \mathbb{R}_{\geq 0}$ be a twice continuously differentiable surrogate loss. For any $x \in \mathcal{M}$, consider a small perturbation $\delta$ decomposed as $\delta = \delta_T + \delta_N$, where $\delta_T \in T_x\mathcal{M}$ and $\delta_N \in N_x\mathcal{M}$, and $\|\delta\| \leq \varepsilon$ for sufficiently small $\varepsilon > 0$.*

 1. *(Margin) There exists $\tau > 0$ such that $|f_\theta(x)| \geq \tau$ for every $x \in \mathcal{M}$.*

 2. *(Input smoothness) $f_\theta$ is $L_x$-Lipschitz and its gradient is $\beta_x$-Lipschitz on the tubular neighborhood $\mathcal{N}_{\delta_0}(\mathcal{M})$.*

 3. *(Loss smoothness) $\mathcal{L}$ is $L_z$-Lipschitz and its derivative $g(z, y) := \partial\mathcal{L}(z, y)/\partial z$ is $\beta_z$-Lipschitz in $z$.*

*Under the above assumptions, the change in loss due to tangent and normal perturbations can be characterized as:*

$$\mathcal{L}(f_\theta(x + \delta_T), y) - \mathcal{L}(f_\theta(x), y) = \mathcal{O}(\varepsilon^2),$$
$$\mathcal{L}(f_\theta(x + \delta_T + \delta_N), y) - \mathcal{L}(f_\theta(x), y) = g(f_\theta(x), y)\nabla_x f_\theta(x)^\top \delta_N + \mathcal{O}(\varepsilon^2).$$

*Proof.* For fixed $(x, y)$ and perturbation $\delta$, apply multivariate Taylor expansion to the composite function $x \mapsto \mathcal{L}(f_\theta(x), y)$ around $x$:

$$\mathcal{L}\big(f_\theta(x + \delta), y\big) = \mathcal{L}(f_\theta(x), y) + g(f_\theta(x), y)\,\nabla_x f_\theta(x)^\top \delta + R_L(x, \delta), \tag{11}$$

where $R_L$ denotes the second-order remainder term.

We now claim

$$|R_{\mathcal{L}}(x, \delta)| \leq \frac{1}{2}\left(\beta_z L_x^2 + L_z \beta_x\right)\|\delta\|^2.$$

Define $h_\theta(x) := \mathcal{L}(f_\theta(x), y)$. Since both $f_\theta$ and $\mathcal{L}$ are twice continuously differentiable, the Hessian of the composition $h_\theta = \mathcal{L} \circ f$ satisfies the chain rule:

$$\nabla_x^2 h_\theta(x) = g'\big(f_\theta(x), y\big)\,\nabla_x f_\theta(x)\,\nabla_x f_\theta(x)^\top + g\big(f_\theta(x), y\big)\,\nabla_x^2 f_\theta(x), \tag{12}$$

where $g(z, y) = \partial\mathcal{L}/\partial z$ and $g'(z, y) = \partial^2\mathcal{L}/\partial z^2$.

From Assumption (A3), we know that $|g(z, y)| \leq L_z$ and $|g'(z, y)| \leq \beta_z$ for all $z$. Assumption (A2) gives $\|\nabla_x f_\theta(x)\| \leq L_x$ and $\|\nabla_x^2 f_\theta(x)\| \leq \beta_x$. Applying submultiplicativity of the operator norm and using $\|uv^\top\| = \|u\|\|v\|$, we obtain:

$$\left\|\nabla_x^2 h_\theta(x)\right\| \leq \beta_z L_x^2 + L_z \beta_x. \tag{13}$$

From Taylor's theorem, there exists $\xi \in (0, 1)$ such that:

$$R_{\mathcal{L}}(x, \delta) = \frac{1}{2}\delta^{\top}\nabla_x^2 h_{\theta}(x + \xi\delta)\delta.$$

Taking norms and applying the bound in equation 13, we get:

$$|R_{\mathcal{L}}(x, \delta)| \leq \frac{1}{2}\|\nabla_x^2 h_{\theta}(x + \xi\delta)\| \cdot \|\delta\|^2 \leq \frac{1}{2}\left(\beta_z L_x^2 + L_z \beta_x\right)\|\delta\|^2.$$

Now decompose the admissible perturbation $\delta$ orthogonally as:

$$\delta = \delta_T + \delta_N, \qquad \delta_T \in T_x\mathcal{M}, \ \delta_N \in N_x\mathcal{M}, \qquad \|\delta\| \leq \varepsilon.$$

By Lemma 1, we have $\nabla_x f_{\theta}(x)^{\top}\delta_T = 0$. Hence, the linear term in the Taylor expansion equation 11 vanishes for $\delta_T$, and we obtain:

$$\mathcal{L}(f_{\theta}(x + \delta_T), y) - \mathcal{L}(f_{\theta}(x), y) = \mathcal{O}(\varepsilon^2).$$

Now consider the full perturbation $\delta = \delta_T + \delta_N$. The total change in loss becomes:

$$\mathcal{L}(f_{\theta}(x + \delta), y) - \mathcal{L}(f_{\theta}(x), y) = g(f_{\theta}(x), y)\nabla_x f_{\theta}(x)^{\top}\delta_N + \mathcal{O}(\varepsilon^2).$$

This concludes the proof.

□

## B   Related Work

This section provides an overview of recent advances in adversarial training that address the trade-off between clean accuracy and robustness. These methods are often motivated by the observation that deep neural networks, even when over-parameterized, suffer from limited effective capacity during adversarial training. Zhang et al. (2021) empirically demonstrated that such models experience an excessive smoothing effect, which hinders their ability to fit both clean and adversarial data. To address this, various approaches have been proposed to make more efficient use of this limited capacity by treating individual training samples differently. These methods can broadly be categorized into two types: (1) reweighting the loss based on data difficulty, and (2) assigning adaptive perturbation bounds tailored to each sample.

One line of work (Zhang et al., 2021; Wang et al., 2021; 2020; Kim et al., 2021) focuses on reweighting the loss function by assigning larger weights to important data points. Zhang et al. (2021) (GAIRAT) and Wang et al. (2021) (MAIL) observed that data points with smaller margins, i.e., the distance from a data point to the model's decision boundary, are more prone to adversarial attacks. GAIRAT estimated the margin by counting the number of iterations in the projected descent method required to generate a misclassified adversarial example, while MAIL introduced the probabilistic margin which leverages the posterior probabilities of classes. Both methods then reweighted the loss function by giving greater importance to data points with smaller margins. Wang et al. (2020) (MART) suggested that misclassified examples are crucial for enhancing robustness and therefore introduced additional regularization on such examples. Kim et al. (2021) (EWAT) reweighted the loss according to the entropy of the predicted distribution.

Another line of work (Ding et al., 2020; Cheng et al., 2022; Balaji et al., 2019) that treats data unequally provides an adaptive perturbation bound instead of a fixed one. Ding et al. (2020) (MMA) identified the margin for each data at each training epoch and utilized it as its perturbation magnitude. This allowed MMA to achieve margin maximization and improve the model's adversarial robustness. Cheng et al. (2022) (CAT) determined a non-uniform effective perturbation length by utilizing the customized training labels. Although not explicitly mentioned, Zhang et al. (2020) (FAT) also employed adaptive perturbation bounds through training on friendly adversarial data that minimizes classification loss among the certainly misclassified adversarial examples. Driven by the motivation that using a common perturbation level may be inefficient and suboptimal, we also adopt varying perturbation bounds in this paper. However, in contrast to most margin-based studies, our approach does not directly rely on the margin. Instead, its tangent space perspective naturally prevents the undesired margin increase. See Appendix D for more details.

## C Projection Matrix

For a matrix $\boldsymbol{A}$ of size $n \times k$ with full column rank, we define $\mathcal{C}_{\boldsymbol{A}}$ as the column space of $\boldsymbol{A}$, which is a linear subspace in $\mathbb{R}^n$ spanned by the columns of $\boldsymbol{A}$. The column space $\mathcal{C}_{\boldsymbol{A}}$ can be represented as:

$$\begin{aligned}
\mathcal{C}_{\boldsymbol{A}} &= \{\beta_1 \boldsymbol{u}_1 + \beta_2 \boldsymbol{u}_2 + \cdots \beta_k \boldsymbol{u}_k : \beta_1, \beta_2, \cdots, \beta_k \in \mathbb{R}\} \\
&= \{\boldsymbol{A}\boldsymbol{\beta} : \boldsymbol{\beta} \in \mathbb{R}^k\}
\end{aligned}$$

where $\boldsymbol{u}_1, \boldsymbol{u}_2, \cdots, \boldsymbol{u}_k$ are the columns of $\boldsymbol{A}$.

The projection matrix $\Pi_{\boldsymbol{A}} := \boldsymbol{A}(\boldsymbol{A}^\intercal \boldsymbol{A})^{-1} \boldsymbol{A}^\intercal$ is an $n \times n$ matrix that defines the linear operator projecting any vector $\boldsymbol{v} \in \mathbb{R}^n$ onto the subspace $\mathcal{C}_{\boldsymbol{A}}$. To derive this matrix, we use the fact that the residual vector $\boldsymbol{v} - \Pi_{\boldsymbol{A}}\boldsymbol{v}$ must be orthogonal to $\mathcal{C}_{\boldsymbol{A}}$. In addition, there exists a vector $\boldsymbol{\beta} \in \mathbb{R}^k$ such that $\Pi_{\boldsymbol{A}}\boldsymbol{v} = \boldsymbol{A}\boldsymbol{\beta}$ since $\Pi_{\boldsymbol{A}}\boldsymbol{v}$ lies in the column space of $\boldsymbol{A}$. Therefore,

$$\begin{aligned}
\boldsymbol{v} - \Pi_{\boldsymbol{A}}\boldsymbol{v} = \boldsymbol{v} - \boldsymbol{A}\boldsymbol{\beta} &\perp \mathcal{C}_{\boldsymbol{A}} \\
\Leftrightarrow \boldsymbol{A}^\intercal(\boldsymbol{v} - \boldsymbol{A}\boldsymbol{\beta}) &= 0 \\
\Leftrightarrow \boldsymbol{\beta} &= (\boldsymbol{A}^\intercal \boldsymbol{A})^{-1} \boldsymbol{A}^\intercal \boldsymbol{v} \\
\Leftrightarrow \Pi_{\boldsymbol{A}} &= \boldsymbol{A}(\boldsymbol{A}^\intercal \boldsymbol{A})^{-1} \boldsymbol{A}^\intercal.
\end{aligned}$$

It is important to highlight that $\boldsymbol{A}^\intercal \boldsymbol{A}$ is always invertible when $\boldsymbol{A}$ has full column rank, ensuring that the projection matrix $\Pi_{\boldsymbol{A}}$ is well-defined.

## D Connection between TART and Margin

Rade & Moosavi-Dezfooli (2022) demonstrated that adversarial training results in an excessive increase in the margin along certain adversarial directions, which leads to a decrease in clean accuracy. While our approach does not directly optimize for the margin, its tangent space perspective provides an inherent mechanism for preventing this undesired margin increase. This is because TART refrains from training on large normal components. We illustrate this by utilizing a toy example proposed by Rade & Moosavi-Dezfooli (2022). This example is a 3D binary classification problem where two classes lie in two noisy concentric circles parallel to the $x_1 x_2$-plane but are separable along the $x_3$ dimension.

As shown in Figure 4, we use clean training, standard AT, and TART to train a single hidden layer multilayer perceptron and compare the resulting decision boundaries. Standard AT achieves robustness at the expense of an unbounded margin in the $x_3$ direction, resulting in a loss of clean accuracy. In contrast, TART maintains a finite margin and improves clean accuracy without compromising robustness. These results not only validate our theoretical insight but also highlight TART's potential to mitigate undesirable geometric side effects of adversarial training. By implicitly regularizing the decision boundary through a tangent-aware perturbation scheme, TART preserves both robustness and generalization, demonstrating a promising direction for geometry-guided defenses.

## E Training Time

We compare the average training time per epoch for five defense methods: standard AT, TART, GAIRAT (Zhang et al., 2021), TRADES (Zhang et al., 2019), and MART (Wang et al., 2020). All models are trained on CIFAR-10 using WideResNet-32-10 for 100 epochs with a single NVIDIA GeForce RTX 3090 GPU. As shown in Table 4, TART takes approximately 1.24 times longer per epoch than standard AT. This marginal increase in training time is mainly due to the additional computation required for estimating the tangent space. The training time of TART is similar to, or slightly shorter than, that of other advanced defense methods such as GAIRAT, TRADES, and MART.

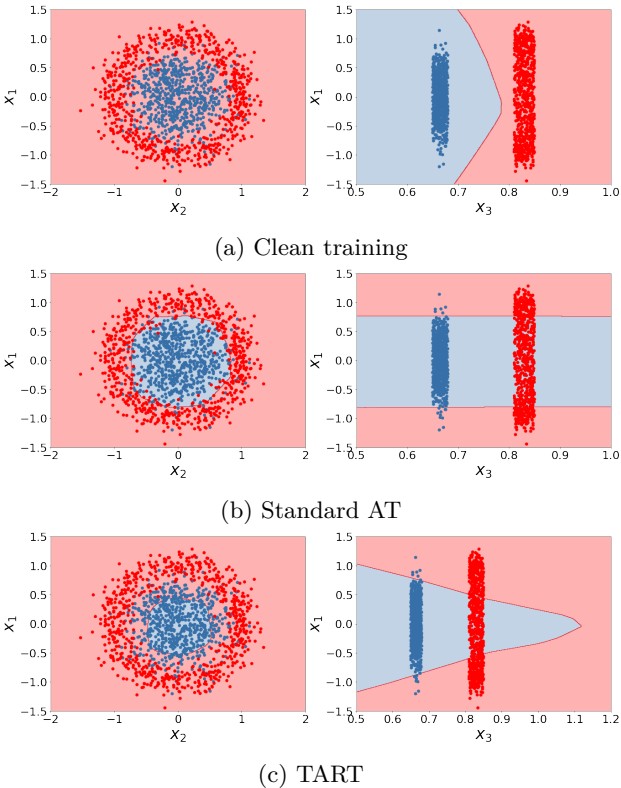

Figure 4: Decision boundary visualization for the toy problem by Rade & Moosavi-Dezfooli (2022): (Left) $x_3 = 0.85$, (Right) $x_2 = 0$. Standard AT considerably enhances robustness from 56% to 77% but results in a reduction in accuracy from 100% to 92%. TART recovers the accuracy to 96% and even slightly improves the robustness to 79%.

| AT | TART | GAIRAT | TRADES | MART |
|---|---|---|---|---|
| 406.12s | 506.66s | 552.36s | 722.02s | 497.46s |

Table 4: Training Time Comparison.

## F Experimental Setup on Tiny ImageNet

Tiny ImageNet (Le & Yang, 2015) is a widely used benchmark in computer vision research. It contains a total of 120,000 images across 200 classes, with 500 training, 50 validation, and 50 test images per class. All images are $64 \times 64$ pixels, making it a more challenging and large-scale dataset than CIFAR-10.

### F.1 Defense Settings

We adopt the VGG-16 architecture (Simonyan & Zisserman, 2014) for all models. Training is performed for 100 epochs using SGD with momentum 0.9 and a weight decay of 0.0002. The initial learning rate is set to 0.1 and is reduced by a factor of 10 at the 60th and 90th epochs. For adversarial training, we apply a 7-step PGD (PGD[7]) attack with a random initialization, an $l_\infty$ perturbation bound of $\epsilon_{\max} = 0.01$, and a step size of $\alpha = 0.002$. No data augmentation is used throughout the training. All experiments are conducted on a single NVIDIA GeForce GTX 1080 GPU.

### F.2 Evaluation Metrics

We evaluate the final model's performance based on clean test accuracy and robustness against three adversarial attacks: Fast Gradient Sign Method (FGSM) (Goodfellow et al., 2014), PGD[20], PGD[40] (Madry et al., 2018), and AutoAttack (Croce & Hein, 2020). All attacks operate within an $l_\infty$ perturbation bound of $\epsilon_{\max} = 0.01$. PGD[20] and PGD[40] use a step size of $\alpha = 0.002$, and FGSM is implemented as a single-step PGD with step size equal to $\epsilon$. Both FGSM and PGD are white-box attacks, assuming full access to the model. AutoAttack combines four diverse attacks to provide a comprehensive robustness assessment.

## G Autoencoder Details

In this section, we describe the autoencoder architecture utilized for all experiments conducted on CIFAR-10 in Section 5.2.

- Encoder:

    1. Convolution layer with $4 \times 4$ filter, # of input channels: 3, # of output channels: 12, stride = 2, padding = 1
    2. ReLU activation layer
    3. Convolution layer with $4 \times 4$ filter, # of input channels: 12, # of output channels: 24, stride = 2, padding = 1
    4. ReLU activation layer
    5. Convolution layer with $4 \times 4$ filter, # of input channels: 24, # of output channels: 48, stride = 2, padding = 1
    6. ReLU activation layer
    7. Convolution layer with $4 \times 4$ filter, # of input channels: 48, # of output channels: 96, stride = 2, padding = 1
    8. ReLU activation layer
    9. Linear layer, # of input channels: 384, # of output channels: 128

- Latent: 128-dimensional

- Decoder:

    1. Linear layer, # of input channels: 128, # of output channels: 384
    2. Transposed-Convolution layers with $4 \times 4$ filter, # of input channels: 96, # of output channels: 48, stride = 2, padding = 1
    3. ReLU activation layer
    4. Transposed-Convolution layers with $4 \times 4$ filter, # of input channels: 48, # of output channels: 24, stride = 2, padding = 1
    5. ReLU activation layer
    6. Transposed-Convolution layers with $4 \times 4$ filter, # of input channels: 24, # of output channels: 12, stride = 2, padding = 1
    7. ReLU activation layer
    8. Transposed-Convolution layers with $4 \times 4$ filter, # of input channels: 12, # of output channels: 3, stride = 2, padding = 1
    9. Sigmoid activation layer

| Defense | Clean (Last) | Clean (Best) | FGSM | PGD$^{20}$ | PGD$^{40}$ | AutoAttack |
|---|---|---|---|---|---|---|
| Clean | $90.72 \pm 0.10$ | $90.86 \pm 0.10$ | $5.19 \pm 1.03$ | $0.0 \pm 0.0$ | $0.0 \pm 0.0$ | $0.0 \pm 0.0$ |
| AT | $81.53 \pm 0.21$ | $81.76 \pm 0.22$ | $54.11 \pm 0.37$ | $37.06 \pm 0.27$ | $36.73 \pm 0.20$ | $35.89 \pm 0.19$ |
| TART (64) | $83.13 \pm 0.33$ | $83.40 \pm 0.19$ | $54.69 \pm 0.16$ | $37.21 \pm 0.14$ | $36.83 \pm 0.12$ | $35.85 \pm 0.01$ |
| TART (128) | $\mathbf{83.47 \pm 0.22}$ | $\mathbf{83.73 \pm 0.15}$ | $\mathbf{55.27 \pm 0.25}$ | $\mathbf{37.72 \pm 0.03}$ | $\mathbf{37.41 \pm 0.13}$ | $\mathbf{36.58 \pm 0.08}$ |

Table 5: Test accuracies (%) on CIFAR-10 with WRN-32-10. The performances over three trials are reported along with their standard deviation.

To investigate how the dimensionality of the latent space affects performance, we conduct experiments on CIFAR-10 using two variants of an autoencoder: one with a 64-dimensional latent space and another with a 128-dimensional latent space. For the 64-dimensional case, we also adjust the architecture accordingly by setting the number of output channels in the encoder's ninth layer and the number of input channels in the decoder's first layer to 64. All other experimental settings, including defense configurations, are identical to those described in Section 5.2.

Table 5 presents the performance comparison of clean training, AT, TART (64), and TART (128), where TART (64) and TART (128) use autoencoders with 64- and 128-dimensional latent spaces, respectively. We observe that even with a low-dimensional latent space (64), TART improves clean accuracy without sacrificing robustness. Moreover, TART (128) achieves even better performance, likely due to its ability to provide a more accurate estimation of the tangent space, resulting in more effective guidance during training.

## H Ablation Study on Perturbation Bound Assignment

In TART, the perturbation bound, $\epsilon$, is adaptively assigned to each data point to improve clean accuracy while maintaining model robustness. Our core approach, TART, employs a straightforward strategy: it allocates a maximum bound of $\epsilon_{\max}$ to the top 50% of data points with the largest tangential components and a bound of 0 to the bottom 50%. The primary advantage of this simple, binary assignment is its significant computational efficiency. By using only two perturbation bounds, 0 and $\epsilon_{\max}$, our method avoids the time-consuming process of regenerating adversarial examples, a factor that accounts for the majority of the training time in many adversarial training methods.

To validate this design choice, we conducted an ablation study on alternative assignment strategies. We considered two main variants:

- TART-$\alpha$: A generalization of our core approach, which assigns a bound of $\epsilon_{\max}$ to the top $\alpha$% of data points and 0 to the remaining $(1 - \alpha)$%.

- TART-prop: A proportional method that assigns the perturbation bound based on the magnitude of the tangential component (TC), given by the formula $\epsilon_i = \frac{TC_i}{\max_j(TC_j)} \cdot \epsilon_{\max}$.

Our experimental results, summarized in Table 6, demonstrate that all three variants (TART (equivalent to TART-50), TART-75, and TART-prop) successfully improve clean accuracy without degrading robustness. While TART-prop also surpasses standard AT (equivalent to TART-100), its clean and robust performance lags behind TART. Furthermore, while TART-75 achieves similar or slightly better robustness than TART, its clean accuracy drops significantly. Given that TART-prop requires additional training time due to its non-binary assignment, our findings confirm that the simple assignment strategy of TART is highly effective while maintaining superior training efficiency.

| Defense | Clean | FGSM | PGD$^{20}$ | AutoAttack |
|---------|-------|------|------------|------------|
| AT | 81.53 | 54.11 | 37.06 | 35.89 |
| TART | 83.47 | 55.27 | 37.72 | 36.58 |
| TART-75 | 82.14 | 54.91 | 37.99 | 36.79 |
| TART-prop | 83.38 | 54.86 | 37.30 | 36.08 |

Table 6: Evaluation of Different Assignment Methods.

