# OpenReview forum: "Improving Clean Accuracy via a Tangent-Space Perspective on Adversarial Training"
_TMLR — Withdrawn by Authors_

### Review · Reviewer_rBw1 · 2025-09-01

**Summary Of Contributions:**

This work proposes a form of adversarial training (TART) that promises to offer an improved robustness--utility trade-off under evasion attacks on classifiers for real-valued data.

The method is motivated by a central theoretical observation in the binary classification setting:
If the training data lies on some lower-dimensional manifold, and certain regularity conditions hold, then:
(1) For some sufficiently small perturbation radius $\varepsilon$, the change in loss along the tangential direction of the manifold is some polynomially small term. (2) The change in loss in the normal direction is larger by an additive linear term.
Based on this theoretical result, the authors argue that optimally chosen adversarial examples should lie off the data manifold. Training on them may thus hurt clean accuracy.

To remedy this issue, the authors propose a three-step procedure (TART) that prevents the generation of adversarial examples that are far off the tangent space.
This procedure (a) computes adversarial examples as usual (e.g., via PGD), (2) estimates their tangential component, and (3) re-computes adversarial examples with a budget chosen based on the tangential component.

As a concrete implementation for the last two steps, the authors propose (b) pre-training an autoencoder and using a linear approximation of its decoder to estimate the manifold's tangent space, and
(b) setting the adversarial budget for the examples with the bottom-50% tangential magnitude to 0 (along some similarly intuitive alternatives mentioned in the appendix).

To demonstrate the benefit of assigning more budget to examples with the largest tangential magnitude, the authors use a synthetic dataset, for which the tangent space of the data manifold can be computed analytically: A random up-projection of a hemisphere.
On this dataset, they benchmark against a modified form of their algorithm that reduces budget for examples with the smallest tangential magnitude, which performs strictly worse.
Finally, the real-world usefulness of the method is demonstrated on two standard vision datsets (CIFAR-10 and Tiny ImageNet) with standard vision architectures. The proposed method offers higher clean accuracy (1-3 p.p.) and slightly higher perturbed accuracy (< 1 p.p.), even when combined with more recent adversarial training methods.


## Strengths
* The high-level narrative flow is very strong (nicely builds up from background/related work to the main theoretical results and motivates method from there)
* The low-level narrative (e.g., in the abstract, which mirrors the overall paper structure) is also very good
* In general, the quality of the writing, visualizations etc. is high
* The authors provide a very effective, intuitive explanation of their main theoretical result
* Not explicitly discussed in the paper, but: The proposed method is plug-and-play compatible with various other adversarial training methods, and can always recover the underlying adversarial training procedure by letting $\varepsilon_\mathrm{adaptive} = \varepsilon$. So it is always guaranteed to be at least as good for some appropriately chosen parameterization.
* Limitations are very transparently discussed (extra cost of estimating tangent space, projecting to tangent space, re-computing adversarial examples etc.)
* Appropriately chosen procedures are proposed to ameliorate these issues (e.g., pre-computing the tangent space estimates etc.). Their limitations are, again, discussed very transparently (data augmentaiton cannot be used etc.)
* Most assumptions made in the main Theorem appear very weak (Lipschitzness etc.) and should thus apply to a wide range of commonly used binary classifiers.

## Major Weaknesses
* While not obvious, the margin assumption in Theorem 3.1 seems to constrain the model to a trivial class of binary classifiers (see section on evidence below)
* There is a a somewhat significant gap between Theorem 3.1 and the method developed from it. The authors argue as follows: Adversarial examples off manifold --> model behavior poorly defined --> high loss --> large gradients --> new decision boundary that generalizes poorly. While this argument is intuitive, none of the claims therein are substantiated via proofs or references to prior work.
* The method, as presented, appears to be limited to $\ell_\infty$ attacks (see end of Section 2.2). It is not clear why this is the case and which concrete steps would be needed to generalize it to arbitrary $\ell_p$. (To me, it appears like TART should work for any $\ell_p$ norm, so the end of Section 2.2 is somewhat confusing to me.)

## Minor weaknesses
* The modified adversarial training approaches (TRADES, MART, GAIRAT) are not explained
* How to combine these training approaches with TART is not explained (an Algorithm environment in the Appendix might be helpful)
* The first paragraph of the introduction is incredibly generic and can be found in identical form in hundreds of other papers. I do not see this as a major issue, but believe that the paper would be stronger if it started with paragraph 2.
* The projection operation in Eq. 3 is not clearly defined. There are various possible ways to map into an $\ell_p$ ball.
* Section 2.2 mentions "other previous studies" but none are cited
* Section 4.0 introduces TART as a general procedure. But Algorithm 1 only shows the specific instantiation proposed in Section 4.3. It might be better to disentangle the two.
* The additional computational overhead from training an autoencoder is not mentioned as a limitation.
* The experimental evaluation makes very broad statements (e.g., "clean accuracy is consistently boosted") and refers to the Tables to substantiate these statements. Readability could be improved by including key quantitative results ("on average, clean accuracy improves by X p.p.") directly in the main text.
* In Section 5.1, it is not clear how the label for the synthetic data is chosen based on "where z is located".

**Additional Comments:**

## Conclusion
Overall I believe that this is a well-written work with a strong narrative arc. It provides convincing empirical evidence and would be of interest to the broader trustworthy ML community.
My main concern is that (1) not all claims in the chain of motivating arguments are substantiated and (2) the main theoretical argument starting the chain may only apply to a trivial class of models, i.e., does not provide substanstive evidence either.

**Audience:**

Yes

**Audience Explanation:**

Adversarial examples have gained renewed attention following the popularization of.
Adversarial training is still the go-to method for improving adversarial robustness.
So I am extremely confident that any improvement to this widely adopted method would be of interest to at least one reader from the trustworthiness / robustness community.

**Claims And Evidence:**

No

**Claims Explanation:**

## Supported

### Effectiveness of the proposed method
The effectiveness is clearly demonstrated using a well-suited synthetic dataset and appropriately chosen computer vision benchmarks (see strengths above).

### Main theoretical result
I did not check every individual equation for typos, but the overall proof strategy appears sound.

## Not supported

### Implications of the main theoretical result
As mentioned under "major weaknesses" above, the paragraph following the bullet points after Theorem 3.1 is a reasonable, but highly speculative and hand-wavy series of arguments that is neither substantiated through original proofs nor through reference to prior work.

### Applicability of Theorem 3.1
An implicit claim of presenting Theorem 3.1 is that it covers at least some subset of useful models.
However, it appears like the margin assumption $|f_\theta(x)| \geq \tau$ limits the Theorem to binary classifieres that are constant on the entire data manifold, i.e., $\forall x, y \in M: \mathrm{sgn}(f_\theta(x)) = \mathrm{sgn}(f_\theta(y))$.

This can be seen from Lemma 1, which is a direct consequence of the margin assumption. Lemma 1 states that the model's input gradient is orthogonal to the Tangent space. Conversely, this seems to imply that one can move along an arbitrary curve along the manifold without changing the function value.

This issue can also be seen directly from the Theorem's assumptions: Under the margin assumption, the model's sign can only change via a discontinuity where the model's output changes by at least $2 \tau$. However, this discontunuity would immediately contradict the assumption that the model is differentiable. So the model can only be constant on the data manifold.

**I might be overlooking something obvious here, so it would be great if the authors could comment on this.**

**Requested Changes:**

## Necessary changes
* Update the wording in the paragraph following the bullet points in Section 3. Specifically, clarify that the entire argument chain is only plausible speculation. Alternatively, provide sources for each of the individual claims or (which I am aware is likely impossible or out-of-scope for a conference-style paper), provide proofs.
* In particular, remove "These theoretical findings underscore". There is only a single theoretical finding.
* Weaken the assumptions of Theorem 3.1 so that it is applicable to non-constant functions (or explain why it is applicable to non-constant functions during the discussion period)

## Recommended, optional changes
* Resolve "minor weaknesses" above
* Remove "to the best of our knowledge" from abstract. This implies that the literature research was not sufficiently thorough
* "The goal is ..." at the end of Section 2.1: Clarify whether this is the goal of standard or adversarial training
* First sentence below Eq.2: Improve typesetting by not starting sentences with inline math
* Theorem 3.1: Clarify which norm $||\cdot||$ refers to ($\ell_\infty$ or $\ell_2$)
* Theorem 3.1: State that 1-3 are assumptions before listing them, not after. I initially thought they were supposed to be results.
* Theorem 3.1: Clarify that the two $\mathcal{O}(\varepsilon^2)$ are identical.
* Explain why focusing on tangent direction still provides a benefit in terms of robustness (Couldn't an adversary at inference time still decide to craft examples in the normal direciton, which would be completely o.o.d.?)
* Fig 2: Add missing xlabel
* Fig 2 and corresponding text: State how adversarial examples were generated (PGD^k)
* Algorithm 1: Make it clearer what is sampled ($z + \delta e_i$ with $\delta \simeq P$ for some P)
* Second-to-last paragraph in Section 4.2: "Training time only increased by a factor of 1.24" is not informative without mentioning the dataset, model etc. Either ove more information to the main text, or move more information to the appendix
* Table 1: Clarify that this is not standard TART (using autoencoder-based tangent estimation) but ANALYTICAL-TART or similar
* Another nice ablation for Section 4.1 would be to randomly set adaptive budgets to $0$ ("Random-TART"), instead of those with the smallest tangential component (Reverse-TART)

---

### Review · Reviewer_Ed9A · 2025-09-03

**Summary Of Contributions:**

The authors propose a method for robust image classification (dubbed TART) which leads to smaller degradation in clean accuracy as compared to existing robustness methods. Their method is based on an insight that when the clean data lie on a low-dimensional manifold (a common assumption about natural image data), training on adversarial examples with a large normal component to the manifold may negatively impact the classifier's decision boundary, leading to degraded performance. Based on this insight, they propose to adversarially train only on those examples with a large component in the tangent space of the manifold. They show how to implement the required steps in practice where the underlying manifold is not known. Experiments on synthetic data and standard adversarial classification benchmarks show that their method can be flexibly added to existing robust classification methods, obtaining general improvements in both clean and adversarial data.

**Additional Comments:**

I felt that there was a disconnect between the theoretical framing of the method and its practical implementation via the perturbation bound in equation (7). The theory (assuming that it can be fixed) says that a large normal component can lead to large loss, not that a large tangential component does not lead to large loss. If the total size of the perturbation is fixed, then because the normal/tangential components are orthogonal, there will be some correlation here, but it still seems like better alignment with the theory to just select the points with the lowest normal component. Have the authors tried testing this alternative perturbation bound selection method? Is it in fact equivalent to the one proposed?

Another question: Below Table 3 on pg. 9, I think there is a minor error describing the modifications to TART and reverse-TART. Specifically, it is stated to use $\epsilon$ for the largest (resp. smallest) 25% examples and $0$ for the smallest (resp. largest) 25% examples. What is to be done with the middle 50%? I assume the authors meant to use $\epsilon$ for the top 25% and 0 for the bottom 75% for TART and the opposite for reverse-TART, but this should be clarified.

**Audience:**

Yes

**Audience Explanation:**

As the authors explain, it is a known issue that robust image classifiers suffer in terms of clean accuracy. Methods which can obtain a "best of both worlds" (or at least improvements to one of these metrics with a reduced sacrifice for the other) should be of general interest to the community, especially since TART is flexible and should be easy to apply in practice.

**Claims And Evidence:**

No

**Claims Explanation:**

# Theory

I have concerns about the proof of the key Lemma 1. At the bottom of pg. 14, $t$ is effectively defined as $t = -\frac{f_\theta(x)-\tau/2}{\alpha}$. Note that this value of $t$ is *fixed* depending on $x$ and $\tau$, and *cannot be made arbitrarily small*. In particular, it is not clear why the Taylor expansion is still valid for this value of $t$, nor why this value of $t$ should lie in the valid range for the exponential map. As such, the equation just above (9) may not be valid, and especially the first line of pg. 10 ("Since we can shrink $t_0$ so that $|t|<t_0$...") also does not hold. Unless there has been a misunderstanding on my part, I do not see an obvious way to fix the proof.

I believe this may be a fundamental problem with the setup/assumptions for the main lemma. Consider the following case: Suppose that $\mathcal{M}$ is a compact manifold and let $f$ be an *arbitrary* $C^2$ function of the ambient space $\mathbb{R}^d$. Since $f$ is continuous, it obtains a minimum $f_{min}$ on $\mathcal{M}$. Then the function $\tilde{f}(x) = f(x) - f_{min} + \tau$ satisfies $|\tilde{f}(x)| \geq \tau$ on $\mathcal{M}$ (so the margin assumption is met) and has the same smoothness parameters as the original $f$; in particular, it will still satisfy assumption 2 if the original $f$ does. But since $f$ was otherwise arbitrary, it clearly should not need to have a gradient which is orthogonal to the tangent space of $\mathcal{M}$.

Probably there needs to be a stronger assumption connecting the fact that $f_\theta$ is supposed to be a classifier for $\mathcal{M}$ (and is presumably doing so correctly or approximately correctly) for the conclusion of the theorem to hold. I expect that this can be fixed since the result makes intuitive sense, but it will require a significant rework. If there is a misunderstanding on my part, I am happy to be corrected.

# Experiments

The empirical part of the paper is certainly its strong suit at present. TART generally obtains improvements in both adversarial *and* clean accuracy over the basic version of existing robustness baselines, showcasing its versatility. I do not keep up-to-date with the adversarial classification literature, but based on a Deep Research summary of the literature from ChatGPT, the robustness baselines and attack methods appear to be complete and state-of-the-art, with the possible exception that different (and generally larger) base models may be used by the most cutting-edge methods. There were a few omissions from the synthetic data experiment that seem like natural baselines to include and which should be added; see the requested changes below.

# Method

The method makes intuitive sense and the authors provide a number of implementation details to reduce what could otherwise be an unacceptable computation and memory burden.

**Requested Changes:**

1. Please fix the proof of Lemma 1. If I am not mistaken, this will require new assumptions to be made.

2. Please add the standard adversarial training examples to the synthetic data experiment as baselines, rather than just reverse-TART. Please also include the clean accuracy of a standard (non-robust) classifier for the synthetic data (i.e. the "Clean" row included in Tables 2 and 3).

---

### Review · Reviewer_6Tkm · 2025-10-17

**Summary Of Contributions:**

This paper addresses the well-known trade-off in adversarial training, where improving robustness against adversarial attacks often leads to a degradation in the accuracy of the model on clean data. In particular, it introduces a theoretically grounded method called Tangent Direction Guided Adversarial Training (TART), which enhances clean accuracy without sacrificing robustness by leveraging the geometry of the data manifold. The key intuition behind this approach is that adversarial examples with large perturbation components normal to the data manifold are the primary cause of decision boundary distortion and reduced clean accuracy.

The paper first formally proves that the normal component of an adversarial perturbation applied to an instance on the manifold plays the major role in changing the value of the loss function of the model  by pushing the data outside the manifold and inducing an accuracy drop. TART mitigates this issue by first estimating the tangent space of the training images with respect to the manifold to which they belong and then decomposing adversarial perturbations into their tangential and normal components. Based on this decomposition, it adaptively modulates the perturbation bound for each example, effectively training on adversarial instances that are less likely to push the data far off the manifold.

The experimental evaluation on CIFAR-10, TinyImageNet, and the WideResNet-32-10 model supports that TART consistently improves clean accuracy while preserving, and in some cases slightly improving, robustness against adversarial attacks.

**Strengths**
- Provides and proves theoretical results to motivate the intuition behind the approach.
- The approach is intuitive, and the experimental results demonstrate that TART preserves accuracy without degrading robustness compared to other adversarial training methods.
- The paper is clearly written.

**Weaknesses**
- It is not clear how the assumptions of the theorem supporting the intuition translate to real-world scenarios.
- The paper reports a (small) improvement in training time induced by TART over ART, but it does not show the impact on training time of the proposed approach when combined with other adversarial robustness methods, such as TRADES.
- The paper lacks comparisons with other prominent methods that address the accuracy–robustness trade-off.

**Additional Comments:**

I invite the authors to address the following typos and editorial issues:
- Figure (a) is never referenced, why?
- Line 12, Algorithm 2: Is the symbol used to indicate the loss consistent with the one used in the rest of the paper?

**Audience:**

Yes

**Audience Explanation:**

Enhancing the robustness of machine learning models is for sure a relevant and timely research topic. Proposing a method that better addresses the trade-off between robustness and fairness in adversarial training compared to existing approaches could be of great interest to the machine learning security community and potentially set a new state of the art in making machine learning models more robust.

**Broader Impact Concerns:**

No impact concerns to signal.

**Claims And Evidence:**

No

**Claims Explanation:**

The paper does a good job of supporting its main claim that the proposed method improves accuracy without degrading robustness compared to other adversarial training methods. Moreover, it provides a theoretical result that supports the intuition behind the method, which strengthens the contribution. However, there are three points that make the evidence less accurate and clear:
- Theorem 3.1 incorporates three assumptions, but it is not discussed how these assumptions hold for real-world datasets and models such as CIFAR-10, TinyImageNet, and the model considered in the experimental evaluation. This point should be discussed, or it should be stated as a limitation that assessing these assumptions in real-world scenarios is difficult. While the theoretical result is interesting, its applicability in practice should be clarified.
- Other prominent approaches addressing the accuracy–robustness trade-off have not been discussed or used as baselines in the experimental evaluation, in particular [A] and [B]. The authors should discuss these methods in relation to their proposed approach and include them in the experimental evaluation to show how TART performs compared to them. While the intuition behind the proposed method is sound, a comparison with other methods tackling the same problem is necessary to properly assess the strengths and limitations of the current proposal.
- Finally, no experimental results are provided to support the claim that the computational burden introduced by the proposed method, when combined with other adversarial training techniques beyond ART (e.g., TRADES), is minimized (as mentioned in Section 4.2). Please provide the training time when TART is incorporated into other adversarial training methods to give a complete picture of the additional computational cost introduced.

[A] Jin et. al., Randomized Adversarial Training via Taylor Expansion, in CVPR 2023.

[B] Pang et. al., Robustness and Accuracy Could Be Reconcilable by (Proper) Definition, in ICML 2022.

**Requested Changes:**

**Requested changes for acceptance**
- Please compare your approach, both theoretically and experimentally, with the two prominent approaches mentioned above.
- Present the training time achieved by your method when combined with TRADES, MART, and GAIRAT to highlight the effect of your method even with these approaches.
- Discuss the applicability of the assumptions in Theorem 3.1.

---

### Note · Authors · 2025-10-20

**Comment:**

To the Reviewers,

We sincerely thank all the reviewers for dedicating their valuable time and providing us with thoughtful and insightful comments on our paper, "TART: Boosting Clean Accuracy Through Tangent Direction Guided Adversarial Training".

The feedback we received was excellent and provided our research team with several important directions to further develop and deepen the ideas presented in the paper. In particular, upon reflection and further investigation, we agree that there exists a critical flaw in the theoretical justification intended to support the core idea of our TART method. We estimate that resolving this theoretical shortcoming and establishing a new mathematical framework to rigorously support our empirical findings will require significant additional work. Therefore, we have made the difficult decision to withdraw our submission at this time.

Thank you once again for your constructive and high-quality reviews, which have helped us identify the potential for improvement in our paper. We look forward to submitting a revised and stronger version in the future.

Sincerely,
The Authors

**Withdrawal Confirmation:**

I have read and agree with the venue's withdrawal policy on behalf of myself and my co-authors.